# Preoperative Assessment of Skeletal Muscle Mass and Muscle Quality Using Computed Tomography: Incidence of Sarcopenia in Patients with Intrahepatic Cholangiocarcinoma Selected for Liver Resection

**DOI:** 10.3390/jcm11061530

**Published:** 2022-03-10

**Authors:** Francesco Ardito, Alessandro Coppola, Emanuele Rinninella, Francesco Razionale, Gabriele Pulcini, Davide Carano, Marco Cintoni, Maria Cristina Mele, Brunella Barbaro, Felice Giuliante

**Affiliations:** 1Hepatobiliary Surgery Unit, Fondazione Policlinico Universitario A. Gemelli IRCCS, 00168 Rome, Italy; francesco.ardito@unicatt.it (F.A.); francescorazionale@gmail.com (F.R.); felice.giuliante@unicatt.it (F.G.); 2Department of Translational Medicine and Surgery, Università Cattolica del Sacro Cuore, 00168 Rome, Italy; emanuele.rinninella@unicatt.it (E.R.); mariacristina.mele@unicatt.it (M.C.M.); 3General Surgery, Fondazione Policlinico Universitario Campus Bio-Medico, 00128 Rome, Italy; 4Clinical Nutrition Unit, Fondazione Policlinico Universitario A. Gemelli IRCCS, 00168 Rome, Italy; gabriele.pulcini@guest.policlinicogemelli.it (G.P.); marco.cintoni@gmail.com (M.C.); 5Department of Bioimaging and Radiological Sciences, Fondazione Policlinico Universitario A. Gemelli IRCCS, 00168 Rome, Italy; caranodoom@gmail.com (D.C.); brunella.barbaro@unicatt.it (B.B.)

**Keywords:** intrahepatic cholangiocarcinoma, sarcopenia, preoperative risk factor, liver surgery, liver resection, skeletal muscle mass

## Abstract

Background: Sarcopenia is considered a predictor of poor postoperative and long-term results following liver resection for intrahepatic cholangiocarcinoma (ICC). The aim of our study was to assess the incidence of sarcopenia in patients resected for ICC and its relation to preoperative clinical factors. Methods: Patients resected for ICC in our unit, with available preoperative CT scans within one month before operation, were enrolled in the study. Skeletal muscle index (SMI) and skeletal muscle radiodensity (SMD) were assessed for each patient. Results: Thirty patients matched all inclusion criteria. Low SMI values were documented in 15 patients (50.0%), and low SMD values were documented in 10 patients (33.3%). SMI was significantly greater in males (*p* < 0.001). In patients who were underweight, the incidence of low SMI was significantly higher than that of high SMI (*p* = 0.031). In patients who were overweight/obese, the incidence of high SMI was significantly higher than that of low SMI (*p* = 0.003) and the incidence of low SMD was significantly higher than that of high SMD (*p* = 0.038). In the univariate analysis, no preoperative factors (clinical and tumor-related factors), in particular BMI, were found to be independent predictors of low SMI. Conclusions: The incidence of sarcopenia was 50.0% in patients selected for liver resection for ICC and was not related to the preoperative clinical factors. A multidisciplinary evaluation of the nutritional status is fundamental before liver resection in patients.

## 1. Introduction

Intrahepatic cholangiocarcinoma (ICC) represents the second most common primary malignancy of the liver [1]. In recent decades, the incidence of this cancer has grown mainly because of increasing risk factors such as diabetes and obesity [2]. Unfortunately, only one-third of patients have access to liver resective surgery or, more recently, liver transplants, which are the only strategies with a curative intent. However, in these selected patients, the 5-year overall survival rates after radical surgical treatments range between 25% and 40% [3]. Several prognostic factors and models have been evaluated in the literature to predict postoperative outcome following liver resection for ICC. The risk factors are generally related to the type and extension of the liver resection and to the general conditions of the patients. Elevated levels of carbohydrate antigen 19.9 (Ca-19.9), a poorly differentiated tumor, or microvascular invasion represent well-known prognostic factors for the survival of these patients. The prognostic nutritional index and inflammatory levels have been reported to be independent prognostic factors of patients with ICC [4].

Recently, sarcopenia has been considered a prognostic biomarker able to predict poor treatments and survival outcomes in patients with gastrointestinal, hepatobiliary and pancreatic cancers [5]. Sarcopenia is a complex syndrome characterized by progressive, generalized loss of skeletal muscle mass and strength due to multiple possible causes such as malignancy. The role of sarcopenia in impairing prognosis after several types of indications to liver resection, including hepatocellular carcinoma, colorectal metastases, and liver transplantation, was already reported [6]. In 2013, Dodson and colleagues published the first experience in the literature that describes sarcopenia as an independent predictor of mortality following intra-arterial therapy for ICC [7]. After that, several papers reported the relation among sarcopenia, worse surgical outcomes, and poor oncological results in terms of disease-free and overall survival [6,8,9,10,11].

Most of these studies reported the experiences of Eastern hepatobiliary surgery centers.

The aim of our study was to retrospectively evaluate the incidence of sarcopenia in patients resected for ICC in our tertiary referral hepatobiliary surgery center and to assess which preoperative factors could be related to the presence of sarcopenia.

## 2. Materials and Methods

### 2.1. Study Design

This is an observational retrospective study using clinical data in combination with computed tomography (CT) images. The data were extracted from a prospectively collected database established in our unit in January 1987 for all consecutive admissions related to possible liver resection. The inclusion criteria were as follows: (a) patients who underwent liver resection in our unit for histologically proven ICC between January 2010 and December 2019, and (b) the availability of preoperative CT scan images that had been performed in our center within one month before operation.

The perioperative nutritional evaluation (the NutriCatt protocol) is part of our ERAS program approved by the Ethics Committee of the Catholic University of the Sacred Heart (Prot. n. 33896/16; ID: 1326), and previously described [12].

For each patient, the following data were collected: demographic information, preoperative height and weight, levels of body mass index (BMI) that were divided into four categories (<20.0 kg/m^2^, underweight; 20.0 to 24.9 kg/m^2^, normal weight; 25.0 to 29.9 kg/m^2^, overweight; and ≥30 kg/m^2^, obese), preoperative albumin level, liver function tests, and the administration of preoperative chemotherapy. The final pathologic data included the presence of liver cirrhosis, the size and number of tumors, the presence of satellite nodules, tumor grading, the presence of perineural invasion, the presence of microvascular invasion, and tumor stage according to the TNM classification 8th edition [13].

### 2.2. Image Analysis

A body composition analysis was performed on a single CT-Scan slice (DICOM image format) at the level of the third lumbar vertebra (L3), using specific software (Slice-O-Matic v5.0, Tomovision^®^, Montreal, QC, Canada). An image analysis was performed by two investigators with imaging experience and blinded to outcomes to minimize the introduction of bias. Cross-sectional area of skeletal muscle (SMA), subcutaneous adipose tissue (SAT), visceral adipose tissue (VAT), and intermuscular adipose tissue (IMAT) were analyzed on the basis of pre-established thresholds of Hounsfield Units (HU): SMA −29 to 150, SAT −190 to −30, VAT −150 to −50, and IMAT −190 to −30 [14]. Skeletal muscle index (SMI) was calculated by normalizing SMA for the height squared (cm^2^/h^2^), skeletal muscle radiodensity (SMD) was retrieved by finding the mean of the HU of SMA, and the SAT/VAT ratio was obtained by dividing the SAT by the VAT area.

### 2.3. Statistical Analysis

Continuous variables were reported as medians. Categorical variables were expressed in numbers and percentages. The χ^2^ test was used for comparing categorical variables. As reported in a previous study [9], the cut-off values for SMI were 52.5 cm^2^/m^2^ in males and 41.2 cm^2^/m^2^ in females. The cut-off values for SMD were 38.3 HU in males and 31.0 HU in females [9].

A logistic regression analysis was used to determine independent predictors of low SMI and low SMD. In the logistic regression analysis, BMI levels were grouped into three categories: <20.0 kg/m^2^, underweight; 20.0 to 24.9 kg/m^2^, normal weight; and ≥25.0 kg/m^2^, overweight/obese. A preliminary univariable model was created. All of the variables at *p* < 0.2 were used for constructing the multivariable model. In all of the analyses, a *p* < 0.05 was considered statistically significant. Analyses were carried out with SPSS 23.0 software (IBM Corp, Armonk, NY, USA).

Postoperative complications were scored according to the Clavien–Dindo grading system [15]. Major complications were defined as grade ≥3.

The overall survival (OS) was calculated from the date of liver resection until the date of death or censored at last follow up. Survival curves were generated using the Kaplan–Meier method and compared with the log-rank test. Analyses were carried out with SPSS 23.0 Software (SPSS Inc., Chicago, IL, USA).

## 3. Results

Between January 2010 and December 2019, 82 liver resections for ICC were performed at our unit in 74 patients (6 re-resections and 1 third resection). In 30 of these patients, preoperative CT scans were performed in our center within one month before operation, and they are the object of our study. All 30 patients enrolled in the study underwent first liver resection.

The characteristics of the patients are summarized in Table 1.

The mean age was 65 ± 12 years (median 66 years; range 25–79). Six patients were obese (20.0%), and four patients (13.3%) were underweight.

The median SMI value was 47.2 cm^2^/m^2^ (30.1–63.0). SMI was significantly greater in males than in females (mean 52.9 ± 7.4 cm^2^/m^2^ vs. 37.9 ± 6.5 cm^2^/m^2^, respectively; *p* < 0.001).

Low SMI values were documented in 15 patients (50.0%): 9 males and 6 females.

The median SMD was 39.5 HU (23.7–66.8). SMD was not significantly different between males and females (mean 40.9 ± 9.5 HU vs. 39.2 ± 6.4 HU, respectively; *p* = 0.611).

Low SMD values were documented in 10 patients (33.3%): 9 males and 1 female.

### 3.1. Correlation between BMI, SMI, and SMD

Low SMI was documented in resected patients, regardless of the BMI level (Figure 1): in all four patients who were underweight, in 70.0% of the patients with normal weight (7 patients), and in 25.0% of the patients who were overweight/obese (4 patients).

Low SMD was documented in 25.0% of patients who were underweight (1 patient), in 10.0% of the patients who were of normal weight (1 patient), and in 50.0% of patients who were overweight/obese patients (8 patients).

In patients who were underweight (with BMI <20), the incidence of low SMI was significantly higher than the incidence of high SMI (26.7% vs. 0, respectively; *p* = 0.031) (Table 2).

On the other hand, in patients who were overweight/obese (with BMI > 25), the incidence of high SMI was significantly higher than the incidence of low SMI (80.0% vs. 26.7%, respectively; *p* = 0.003) (Table 2). Moreover, in patients who were overweight/obese, the incidence of low SMD was significantly higher than the incidence of high SMD (80.0% vs. 40.0%, respectively; *p* = 0.038) (Table 2).

### 3.2. Correlation between Tumor-Related Factors, SMI, and SMD

SMI and SMD were not correlated with tumor-related factors, including number and size of tumor, grading, the presence of satellite nodules, perineural invasion, and microvascular invasion. Moreover, the TNM stage did not impact the variation in SMI and SMD values.

### 3.3. Predictors of Low SMI and Low SMD: Univariate Analysis

In the univariate analysis, no preoperative factors (clinical and tumor-related factors) were found to be independent predictors of low SMI (Table 3). In particular, BMI was not an independent predictor for low SMI.

### 3.4. Operative Results

Nineteen patients (63.3%) underwent major liver resection (resection of ≥3 liver segments). The 90-day postoperative mortality rate was 0.

Postoperative complications occurred in 11 patients (36.7%). The incidence of complications in patients with low SMI was higher than in patients with high SMI, but the difference did not reach a statistical significance (46.7% vs. 26.7%, *p* = 0.255, respectively).

Major complications occurred in five patients (16.7%). The incidence of major complications in patients with low SMI was higher than in patients with high SMI, but the difference did not reach a statistical significance (20.0% vs. 13.3%, *p* = 0.624, respectively).

### 3.5. Overall Survival

After a mean follow-up of 37 ± 30 months, the 5-year OS of the whole series was 59.8% (median survival 67 months). The five-year OS was lower in patients with low SMI than in patients with high SMI, but the difference was not statistical significant (51.4% vs. 68.6%, *p* = 0.549, respectively).

## 4. Discussion

This study showed that the incidence of sarcopenia in highly selected patients undergoing liver resection for ICC at a tertiary referral hepatobiliary surgery Western center was not negligible. Indeed, out of the 30 patients retrospectively evaluated by a body composition analysis on CT scan, 15 (50.0%) presented with a low value of the skeletal muscle index and were classified as sarcopenic patients.

In the literature, there are a few papers focusing on the incidence of sarcopenia in patients selected for liver resection for ICC, and most of them are from Eastern centers [6,9]. Yugawa and colleagues [6] reported the incidence of sarcopenia as 47.6% in men and as 52.6% in women. In the study by Okumura et al. [9], the rate of sarcopenic patients with low SMI in 109 resected patients with ICC was 63.3%.

Sarcopenia was described in 1989 as a decrease in muscle mass related to age [16]. However, it has been showed that sarcopenia not only is a condition seen in people who are older but it can be observed also in malignant disease associated with malnutrition [17]. The features of sarcopenia include a decreased muscle mass that can be measured by the skeletal muscle index (SMI) using a CT scan and the infiltration of muscle by fat. The increased intramuscular adipose tissue may cause a decreased skeletal muscle quality. This feature may be analyzed in the skeletal muscle radiodensity (SMD) using a CT scan.

Several studies have assessed the impact of sarcopenia by using CT scan, on the postoperative results and the oncologic long-term results following liver resection for ICC [5,6,11,18,19,20,21]. It has been shown that low skeletal muscle mass and low muscle quality were independent risk factors for postoperative mortality following liver resection for ICC [5,6,11,18,19,20,21]. Furthermore, these radiologic features were correlated with significantly lower 5-year overall and disease-free survival rates [5,6,11,18,19,20,21]. In our study, the evaluation of the impact of sarcopenia on postoperative results and oncologic long-term results following liver resection for ICC was not the primary end-point due to the small number of patients enrolled. However, our results confirmed that there was a tendency toward a higher incidence of postoperative complications and of major complications in sarcopenic patients with low SMI. Moreover, the 5-year OS was lower in sarcopenic patients with low SMI than that observed in patients with high SMI, without reaching a statistical significant difference.

Some authors investigated the role of prognostic nutritional markers with sarcopenia. Li and colleagues [4] explored the prognostic value of the preoperative albumin–globulin score with the skeletal muscle index as well as their combination in patients with ICC treated with surgical resection. This combination was related to the postoperative long-term outcomes of surgically treated ICC patients. The authors concluded that their index, which can be preoperatively used, could represent a useful tool for risk classification and for clinical therapeutic decision-making for ICC patients [4]. A similar experience was reported by Yu [10], who showed that patients with high fibrinogen–albumin ratios and incidence of sarcopenia, had shorter overall survival and recurrence-free survival than other patients.

However, most of these studies come from Eastern hepatobiliary surgery centers [6,8,9,10,18,19]. The aim of our study was to retrospectively evaluate the incidence of sarcopenia in patients with ICC who were highly selected for liver resection at our Western center. Our results showed that a low skeletal muscle index was preoperatively documented in half of resected patients. In other words, although accurate preoperative clinical selection was performed in all patients, indeed the preoperative albumin levels were normal in 96.7% of patients, and only 4 patients (13.3%) presented underweight, the incidence of sarcopenia was high (50.0%).

We also evaluated the relation between BMI and sarcopenia. It has been demonstrated that obesity may be a risk factor for cancer [22] and may be associated with sarcopenia. In our series, 53.3% of patients were overweight/obese. This result is completely different from those reported in other Eastern series. Indeed, in a recent paper by Okumura et al. [9], 81.7% of patients who underwent liver resection for ICC were not overweight (BMI < 25 kg/m^2^). In our series, low SMI was significantly more frequent than high SMI in patients who were underweight. On the other hand, in patients who were overweight/obese, high SMI was significantly more frequent than low SMI. Moreover, in patients who were overweight/obese, low SMD was significantly more frequent than high SMD. However, in the univariate analysis, BMI was not an independent predictor for low SMI. These results confirm that sarcopenia cannot be preoperatively suspected only with BMI evaluation. If in our series we exclude patients who were underweight and overweight where sarcopenia may be suspected, 46.7% of the low SMI cases were documented in the group of patients who were of normal weight.

In this study 50.0% of patients presented with tumors ≥ 5 cm and 30.0% of patients had multinodular disease. T stage 3–4 was documented in 26.7% of cases, and 26.7% of patients had lymph node metastases. It was interesting to note that SMI and SMD were not correlated with tumor-related factors, including number and size of tumor, grading, the presence of satellite nodules, perineural invasion, and microvascular invasion. Moreover, the TNM stage did not impact on the variation in SMI and SMD values. In the univariate analysis, none of the tumor-related factors associated with advanced ICC were an independent predictor for low SMI.

This study had several limitations. Indeed, this is a retrospective study performed at a single center, which collected a small number of patients. However, this study highlighted the fundamental importance of a multidisciplinary preoperative evaluation of patients undergoing liver resection by specialized dietitians. Indeed, in such patients, the evaluation of a single clinical parameter may be not enough to assess the exact nutritional status of the patient in order to exclude the presence of malnutrition and sarcopenia. A complete preoperative evaluation of the nutritional status is fundamental in patients undergoing liver resection. The preoperative identification of malnutrition and nutrition therapy before surgery are two essential steps in decreasing the risk of postoperative complications. In a recent study by our center [12], we demonstrated how a highly specialized preoperative multidisciplinary evaluation of surgical patients via the adoption of a standardized and personalized nutritional protocol (NutriCatt protocol) within the ERAS program may be related to a reduction in the length of hospital stay without any increase in postoperative morbidity and mortality.

## 5. Conclusions

In conclusion, despite the limitations of a retrospective analysis on a small group of cases, our study showed that the incidence of sarcopenia in highly selected patients undergoing liver resection for ICC at a tertiary referral hepatobiliary surgery Western center was not negligible. Moreover, the presence of sarcopenia was not related to clinical factors and tumor-related factors, suggesting that the preoperative identification of malnutrition may not be performed using single clinical parameters, but a complete preoperative evaluation of the nutritional status is fundamental in patients undergoing liver resection.

## Figures and Tables

**Figure 1 jcm-11-01530-f001:**
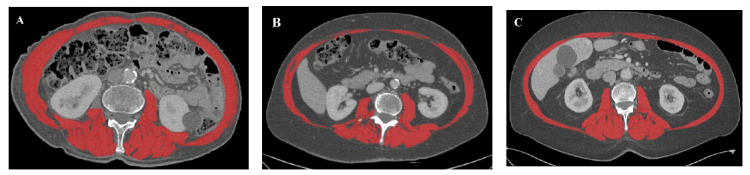
Cross-sectional CT scan at the level of the third lumbar vertebra. Three cases of sarcopenic patients with different BMI are shown: (**A**) a 75-year-old male patient who was underweight (SMI = 46.9 cm^2^/m^2^; BMI = 18.4 kg/m^2^), (**B**) a 73-year-old female patient who was of normal weight (SMI = 30.4 cm^2^/m^2^; BMI = 23.2 kg/m^2^), and (**C**) a 65-year-old male patient who was overweight (SMI = 38.8 cm^2^/m^2^; BMI = 27.8 kg/m^2^).

**Table 1 jcm-11-01530-t001:** Patient demographics.

Characteristics of the 30 Patients Resected for ICC	
Variables	
Age, yr, median (range)	66 (25–79)
Sex, No. (%)	
Male	20 (66.7)
Female	10 (33.3)
BMI, Kg/m^2^, median (range)	25.6 (15.9–36.2)
BMI, No. (%)	
<20.0 kg/m^2^	4 (13.3)
20.0–24.9 kg/m^2^	10 (33.3)
25.0–29.9 kg/m^2^	10 (33.3)
≥30 kg/m^2^	6 (20.0)
Preoperative chemotherapy, No. (%)	2 (6.7)
Presence of cirrhosis, No. (%)	3 (10.0)
Preoperative albumin (g/dL), median (range)	41 (23–48)
Preoperative total bilirubin (mg/dL), median (range)	0.6 (0.3–1.9)
Preoperative protrombin time (%), median (range)	93 (64–281)
Preoperative platelet count (10^9^/L), median (range)	234 (85–389)
Tumor size ≥ 5 cm, No. (%)	15 (50.0)
Tumors number	
Single tumor	21 (70.0)
>1 nodule	9 (30.0)
Satellite nodules, No. (%)	10 (33.3)
Grading	
G 1	18 (60.0)
G 2–3	12 (40.0)
Perineural invasion, No. (%)	13 (43.3)
Microvascular invasion, No. (%)	10 (33.3)
T stage	
T 1–2	22 (73.3)
T 3–4	8 (26.7)
Lymph node metastases, No. (%)	8 (26.7)
SMI, cm^2^/m^2^, median (range)	47.2 (30.1–63.0)
SMD, HU, median (range)	39.5 (23.7–66.8)

**Table 2 jcm-11-01530-t002:** Relation between clinical and tumor-related factors with SMI and SMD.

Variables	Low SMI *n* = 15	High SMI *n* = 15	*p*-Value	Low SMD *n* = 10	High SMD *n* = 20	*p*-Value
Age, yr, median (range)	66 (25–76)	65 (48–79)	0.797	73 (44–79)	69 (25–76)	0.149
Sex, No. (%)			0.438			0.055
Male	9 (60.0)	11 (73.3)		9 (90.0)	11 (55.0)	
Female	6 (40.0)	4 (26.7)		1 (10.0)	9 (45.0)	
BMI, No. (%)						
<20.0	4 (26.7)	0	0.031	1 (10.0)	3 (15.0)	0.704
20.0–24.9	7 (46.7)	3 (20.0)	0.121	1 (10.0)	9 (45.0)	0.055
≥25.0	4 (26.7)	12 (80.0)	0.003	8 (80.0)	8 (40.0)	0.038
Preoperative chemotherapy, No. (%)	2 (13.3)	0	0.143	1 (10.0)	1 (5.0)	0.604
Presence of cirrhosis, No. (%)	0	3 (20.0)	0.067	2 (20.0)	1 (5.0)	0.196
Preoperative albumin < 34 g/dL, No. (%)	0	1 (6.7)	0.309	0	1 (5.0)	0.519
Preoperative total bilirubin (mg/dL), median (range)	0.6 (0.3–1.9)	0.6 (0.4–1.3)	0.767	0.8 (0.4–1.3)	0.6 (03–1.9)	0.496
Preoperative protrombin time (%), median (range)	94 (64–115)	91 (72–117)	0.941	83 (74–105)	97 (64–281)	0.313
Preoperative platelet count (10^9^/L), median (range)	247 (196–389)	210 (114–343)	0.005	202 (122–389)	237 (85–343)	0.884
Tumor size ≥ 5 cm, No. (%)	7 (46.7)	8 (53.3)	0.715	6 (60.0)	9 (45.0)	0.438
Tumors number			0.232			1
Single tumor	9 (60.0)	12 (80.0)		7 (70.0)	14 (70.0)	
>1 nodule	6 (40.0)	3 (20.0)		3 (30.0)	6 (30.0)	
Satellite nodules, No. (%)	6 (40.0)	4 (26.7)	0.438	3 (30.0)	7 (35.0)	0.784
Grading			1			0.429
G 1	9 (60.0)	9 (60.0)		5 (50.0)	13 (65.0)	
G 2–3	6 (40.0)	6 (40.0)		5 (50.0)	7 (35.0)	
Perineural invasion, No. (%)	8 (53.3)	5 (33.3)	0.269	5 (50.0)	8 (40.0)	0.602
Microvascular invasion, No. (%)	5 (33.3)	5 (33.3)	1	5 (50.0)	5 (25.0)	0.170
T stage			1			0.559
T 1–2	11 (73.3)	11 (73.3)		8 (80.0)	14 (70.0)	
T 3–4	4 (26.7)	4 (26.7)		2 (20.0)	6 (30.0)	
Lymph node metastases, No. (%)	4 (26.7)	4 (26.7)	1	4 (40.0)	4 (20.0)	0.242

**Table 3 jcm-11-01530-t003:** Univariate analysis of clinical and tumor-related factors associated with low SMI.

Variables	Univariate Analysis *p*-Value
Age > 70 years	0.713
Male sex	0.441
BMI	0.570
<20.0	0.999
20.0–24.9	0.999
≥25.0	0.999
Preoperative chemotherapy	0.999
Presence of cirrhosis	0.999
Preoperative albumin < 34 g/dL	1
Tumor size ≥ 5 cm	0.715
Tumors number > 1	0.239
Satellite nodules,	0.441
Grading 2–3	1
Perineural invasion	0.335
Microvascular invasion	1
T stage 3–4	1
Lymph node metastases	1

## Data Availability

Not applicable.

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
