# Peer review of "Preoperative Assessment of Skeletal Muscle Mass and Muscle Quality Using Computed Tomography: Incidence of Sarcopenia in Patients with Intrahepatic Cholangiocarcinoma Selected for Liver Resection"

_jcm, 2022, doi:10.3390/jcm11061530_

Round 1

Reviewer 1 Report

The manuscript is very well written and statistically analyzed. However, Necessary biochemical profiles of patients have not been studied like (LFTs)

The discussion part should be improved to clarify the  incidence of sarcopenia in patients resected for ICC and its relation with clinical factors

Author Response

We thank the Reviewer for Her/His  detailed analyses of our manuscript and for Her/His constructive criticisms and advice. We have modified the manuscript according to their comments and questions.

All the new inclusions and changes performed in the text following the comments are highlighted in yellow. 

Comment: The manuscript is very well written and statistically analyzed. However, Necessary biochemical profiles of patients have not been studied like (LFTs)

Response: Preoperative liver function tests have been included in the analysis. In details, total bilirubin levels, prothrombin time and platelet count and their relation with SMI and SMD have been included in tables 1 and 2.

Comment: The discussion part should be improved to clarify the incidence of sarcopenia in patients resected for ICC and its relation with clinical factors.

Response: A comment about the reported incidence of sarcopenia in resected patients for ICC has been included in the Discussion section on page 9 lines 202-206.

Reviewer 2 Report

The authors present a series of patients with ICC and attempt to correlate preoperative factors for CT-measured sarcopenia

1) Did preoperative chemotherapy affect sarcopenia. I realize this is a minority of patients

2) I would exclude the patients that had re-resections and focus on those with primary tumor resection to make the cohort more homogeneous

3) Although not the focus of the manuscript, do the authors have any data on the outcomes of the patients, either in regards to complications or survival?

Author Response

We thank the Reviewer for Her/His detailed analyses of our manuscript and for Her/His constructive criticisms and advices. We have modified the manuscript according to their comments and questions.

All the new inclusions and changes performed in the text following the comments are highlighted in yellow.

Comment: The authors present a series of patients with ICC and attempt to correlate preoperative factors for CT-measured sarcopenia

  • Did preoperative chemotherapy affect sarcopenia. I realize this is a minority of patients.

Response: In the manuscript we analyzed this aspect. However, as reported in Table 1, few patients underwent preoperative chemotherapy (2 patients, 6.7%). In table 2, we analyzed the relation between preoperative chemotherapy and incidence of sarcopenia (low SMI and low SMD): there was not a statistical significant difference.

2) Comment: I would exclude the patients that had re-resections and focus on those with primary tumor resection to make the cohort more homogeneous

Response: In the group of 30 patients, selected for the study, there were not cases of re-resection. We better specified this aspect in the results section on page 3, lines 126-127.

3) Comment: Although not the focus of the manuscript, do the authors have any data on the outcomes of the patients, either in regards to complications or survival?

Response: In our study, the evaluation of the impact of sarcopenia on postoperative results and oncologic long-term results following liver resection for ICC was not the primary end-point due to the small number of patients enrolled. We included a new analysis on postoperative results and long-term results. Details have been included in the methods section on page 3, lines 116-121, in the results section on page 8, lines 177-191 and in the discussion section on page 10 lines 219-226.
